# Environmental enrichment improves social isolation-induced memory impairment: The possible role of ITSN1-Reelin-AMPA receptor signaling pathway

Swamynathan Sowndharya, Koilmani Emmanuvel Rajan○*

Behavioural Neuroscience Laboratory, Department of Animal Science, Bharathidasan University, Tiruchirappalli, India

* emmanuvel1972@yahoo.com

## Abstract

Environmental enrichment (EE) through combination of social and non-biological stimuli enhances activity-dependent synaptic plasticity and improves behavioural performance. Our earlier studies have suggested that EE resilience the stress induced depression/ anxiety-like behaviour in Indian field mice *Mus booduga*. This study was designed to test whether EE reverses the social isolation (SI) induced effect and improve memory. Field-caught mice *M. booduga* were subjected to behaviour test (Direct wild, DW), remaining animals were housed under SI for ten days and then housed for short-term at standard condition (STSC)/ long-term at standard condition (LTSC) or as group in EE cage. Subsequently, we have examined reference, working memory and expression of genes associated with synaptic plasticity. Our analysis have shown that EE reversed SI induced impairment in reference, working memory and other accompanied changes i.e. increased level of Intersectin 1 (ITSN1), Huntingtin (Htt), Synaptotagmin -IV (SYT4), variants of brain-derived neurotrophic factor (Bdnf - III), α-amino-3-hydroxy-5-methyl-4-isoxazolepropionic acid (AMPA) receptor (GluR1) expression, and decreased variants of Bdnf (IV), BDNF, Reelin, Apolipoprotein E receptor 2 (ApoER2), very low-density lipoprotein receptor (VLDLR), Src family tyrosine kinase (SFKs), Disabled protein (Dab)-1, Protein kinase B (PKB/Akt), GluR2, Mitogen-activated protein kinase (MAPK) and Extracellular signal-regulated kinase (ERK1/2) expression. In addition, SI induced reduction in BDNF expressing neurons in dentate gyrus of hippocampus reversed by EE. Further, we found that SI decreases small neuro-active molecules such as Benzenedicarboxylic acid, and increases 2-Pregnene in the hippocampus and feces reversed by EE. Overall, this study demonstrated that EE is effectively reversed the SI induced memory impairment by potentially regulating the molecules associated with the ITSN1-Reelin–AMPA receptor pathway to increase synaptic plasticity.

**Data Availability Statement:** All relevant data are within the paper and its Supporting Information files.

**Funding:** SS is recipient of Council of Scientific and Industrial Research (CSIR)-JRF Research Fellowship (File No. 09/475(0204)/2021-EMR-I). https://newfms.ncl.res.in/ KER received research grant from Department of Science & Technology (EMR/2016/005217 dt.21.03.2018) https://www.serbonline.in/SERB/HomePage The funders to this study had no role in study design, data collection and analysis, decision to publish, or preparation of the manuscript.

**Competing interests:** The authors have declared that no competing interests exist.

**Abbreviations:** EE, Environmental enrichment; STSC, Short-term at standard condition; LTSC, long-term at standard condition; DW, Direct wild; ITSN1, Intersectin-1; Htt, Huntingtin; MAPK, Mitogen-activated protein kinase; BDNF, brain-derived neurotropic factor; REST, Repressor element 1-silencing transcription factor; NRSF, neuron-restrictive silencer factor; NRSE, Neuron-restrictive silencer element; ARMS, ankyrin repeat-rich membrane-spanning; Kidins220, Kinase D-interacting substrate of 220 kDa; SYT-4, Synaptotagmin-IV; ApoER2, Apolipoprotein E receptor 2; VLDLR, Very low density lipoprotein receptor; Dab1, Disabled 1; SFKs, Src family tyrosine kinase; PI3K-PKB/Akt, Phosphoinositide-3-kinase–protein kinase B; AMPA, α-amino-3 hydroxy-5-methyl-4-isoxazole propionic acid; ERK1/2, extracellular signal-regulated kinase; IL-1, interleukin-1; CRH, corticotropin-releasing hormone; ACTH, adreno-corticotropic hormone; SI, social isolation; HBT, Hole-board test; LTP, long-term potentiation; SVs, synaptic vesicles; RRP, readily releasable pool; SNARE, N-ethylmaleimide-sensitive factor attachment protein receptors proteins.

# Introduction

Environmental enrichment (EE) provides a combination of social and non-biological stimuli [1] through the interaction of multi-sensory system, which reduces anxiety/ depressive-like behaviour, and improves learning and memory [2, 3]. The complexity of physical and social stimuli at EE provide potential benefits to hippocampus structure and functions, including the expression of synaptic protein, synaptogenesis and dendritic arborisation [4–7]. It is well known that sustained neurotransmission could positively influence the neuronal communication, i.e. expression of genes associated with synaptic plasticity [8]. The endocytic scaffold protein intersectin-1 (ITSN1) exerts a positive effect on neuronal communications [9] through the activation of endocytosis, and neurotransmitters release [8–10]. Further, mitogen-activated protein kinase (MAPK) dependent signaling pathway facilitate the interaction of ITSN1 with huntingtin (Htt) [11]. Htt has been shown to regulate the transcription of brain-derived neurotropic factor (BDNF) through repressor element-1 silencing transcription factor/ neuron-restrictive silencer factor (REST/NRSF), which bind on the neuron-restrictive silencer element (NRSE) at BDNF promoter [12]. Synthesised BDNF stored in dense core vesicles in pre-synapsis [13], and released by activity dependent calcium influx, ankyrin repeat-rich membrane-spanning/ kinase D-interacting substrate of 220 kDa (ARMS/Kidins220) with synaptotagmin-IV (SYT-4) [14–16]. Synaptically released BDNF positively influence the expression of Synaptophysin (SYP)-1 and activate neuronal signaling [17].

Meanwhile, earlier study has been shown that Reelin an extracellular matrix glycoprotein appears to exert its function by directly binding with the very low-density lipoprotein receptor (VLDLR), apolipoprotein E receptor 2 (ApoER2), and subsequently activate the intracellular adapter protein disabled (Dab)-1 by tyrosine phosphorylation [18]. Phosphorylated Dab-1 then activate Src family tyrosine kinases (SFKs), which subsequently activate other kinases and phosphatases, including Phosphatidylinositol 3-kinase (PI3K)/ protein kinase B (Akt) [19]. Further, Reelin regulate α-amino-3 hydroxy-5-methyl-4-isoxazole propionic acid (AMPA) receptor sub-unit according to the context [20]. Earlier studies demonstrated that through the non-canonical pathway Reelin activate MAPK /extracellular signal-regulated kinase (ERK1/2) depended on SFKs positive feedback mechanism [21]. In addition, intricate interplay between environment and brain trigger diverse action of endogenous neuro-active peptides/ metabolites [22], which could regulate stress hormones/ neurotransmitter and associated signaling pathway.

Environmental enrichment (EE) has been shown to reverses the social isolation (SI) induced alterations in neurotransmitters release/ up-taker [23–25], hippocampal neurogenesis [26, 27], synaptic plasticity [28, 29], anxiety-like behaviour [30, 31] and learning and memory [25, 31–34]. However, the underpinning molecular mechanism needs to be elucidated in different genetic system [35]. Since, field–caught rodents are more sensitive to SI; experiments in field-caught rodent may provide valuable information [36]. Field mice *Mus booduga* live as a group (male, female and pups) in the burrow with first storage chamber and second nesting chamber, occasionally in a single burrow male mice live alone. Since *M. booduga* preferred to live as a social group, housing a single mouse in laboratory cage can be considered as SI stress [37]. Earlier, we have demonstrated that EE reduces anxiety/ depressive-like behaviour in *M. booduga* through the regulation of the stress hormone mediated signalling pathway [2, 3, 38–40]. The present study was designed to test whether EE reverse the effects of SI induced memory impairment and normalize the expression of genes associated synaptic plasticity and memory in field mice *Mus booduga*.

# Materials and methods

## Subject animals

Indian field mice *Mus booduga* were captured with rodent trapper from agricultural field (10° 50′ N; 78°43′ E; Madurai, India) [41] and transferred to Institutional animal maintenance

facility. Male animals (n = 56) were selected based on body weight (Body weight: 9.5 ±1.70g; Age approx.: 3 months), without wound or wound mark anywhere in the body. Animals were maintained under standard laboratory condition (12/12 hr of L/D cycle; 23±1˚C; Rh:50±5%), *ad libitum* of water and food (Pearl millet) were provided except during the experimental session.

## Environmental enrichment

Environmental Enrichment cage (120×100×60 cm) was connected with standard laboratory cage (29.5×22×13cm) through a plastic pipe with bends to mimic the natural burrow system. The cage enriched with different shapes and colors of objects, plastic pipes with bends which acts as a tunnel, running wheels, ladders, balls and mud pot with bedding material (dried and broken paddy straw). The standard laboratory cage serves as a food chamber, where the animals get plenty of food and water [40]. Novelty was maintained in the EE cage by changing the position of objects every three days and introducing new objects by replacing old objects [42].

## Experimental group and housing condition

Flow chart that showing the complete process of an experimental design (S1 Fig in S1 File). On day 2, (i) Direct Wild (DW) group animals (n = 10) were subjected to hole-board test (HBT). Remaining animals (n = 35, 1/cage) were individually housed for social isolation (SI) in small standard laboratory cage (29.5×22×13cm) for ten days [43]. After the period of SI, animals were randomly divided into three groups: (ii) Short-Term at Standard Condition (STSC; n=10), housed at standard laboratory cage (43×27×15cm; 2-3/cage with Kleenex tissue paper as bedding) for seven days and then subjected to HBT; (iii) Long-Term at Standard Conditions (LTSC, n=10, 43×27×15cm; 2-3/cage with Kleenex tissue paper as bedding), which remained in the laboratory cage for another thirty days, then subjected to HBT. (iv) Environmental Enriched (EE) group [n=15 in EE cage] housed for thirty days and then subjected to HBT. At the end of the housing condition, less active animals were excluded from the HBT, animals used for HBT reported as sample size for all groups. Sample size for experimental groups determined by Slovin formula (n = $N/1+Ne^2$) with the margin error of 5% [44]. Immediately after HBT, samples were collected and processed for further analysis.

## Behavioural test

Hole-board Test (HBT) apparatus (44.5 x 44.5 x 30.5 cm) consisted of 16 holes; out of which four holes were baited in a fixed pattern and used to measure the working and reference memory. Animals were food deprived for 8 hrs and transferred to experimental room 1 hr before behavioural test. The entire behaviour test was recorded (Sony HDR-CS405), analysed by the experimenter (DeepLabCut software), who was not aware of the animal group and their treatment. Mice were individually placed on the apparatus for habituation (5 min) with bait on all the holes. Experiment was conducted continuously for four days, Mice were introduced into the apparatus in random position and the test was terminated when the mice visited all four baited holes. The apparatus was cleaned (75% ethanol) after completing every session to clean and remove odor cues. When the mouse touch the hole with the nose was scored as visit or revisit accordingly. The ratio of reference memory (number of visit + revisits to the baited holes / Total No. visits + revisits to baited and non-baited holes) and working memory (No. of food rewarded visits to No. of visits + revisits to baited holes) calculated as described earlier [45].

## Sample preparation

Mice represents different housing conditions were euthanized (sodium pentobarbital 45mg/kg; intraperitonially) after the behavioural test. Hippocampus region (n = 6/group) was dissected out from the whole brain, and stored at -70˚C. The hippocampus tissue from each individual was used to isolate total RNA, protein and Gas Chromatography-Mass Spectrometry (GC-MS) analysis, and feces (n = 6/group) also collected from the same individuals. The whole brain was dissected out (n = 4/group), and immediately processed for immunohistochemistry.

Hippocampus tissue was homogenized in lysis buffer [Tris-Hydrochloride (50mM; pH 7.5; Cat# RM613; HiMedia, India) Sodium chloride (150mM; Cat# 1.93206.0521; Merck, India), Ethylenediaminetetraactetic acid (EDTA) Disodium salt (50 mM; Cat# RM1195; HiMedia, India), Tergitol (NP-40; 0.1% V/V; Cat# MKCD6607; Sigma-Aldrich, India), Dithiotheritol (DTT,1 mM; Cat# MB070; HiMedia, India) Sodium orthovanadate (Na3Vo4, 0.2Mm; Cat# S6508; Sigma-Aldrich, India), Phenylmethylsulfonyl fluoride (PMSF;0.023mM Cat# P7626; Sigma-Aldrich, India)] following the procedure as reported [40]. Final supernatant was collected, concentration of each samples were measured using Bio-photometer Plus (Eppendorf Inc., Germany) and stored in aliquots at -80˚C.

## Western blotting

Total protein (50μg) and protein loading dye [Glycerol (125mM; Cat# 1.94501.0521; Merck, India), Tris-Hydrochloride (pH6.8, Cat#1.94954.0521; Merck India), SDS (4%; Cat# 1.94954.0521, Merck India), Bromophenol Blue (0.006%; Cat# RM117; HiMedia, India), Mercaptoethanol (2%; Cat# MB041; HiMedia, India)] was mixed then resolved on Polyacrylamide gel (PAGE; 10/12%; Cat# 1610182-85, TGX Stain free Acrylamide kit; Bio-Rad Laboratories, Inc., USA). Resolved proteins were transferred onto a membrane (Mini PVDF Transfer packs) with Trans-Blot Turbo Transfer System (Bio-Rad laboratories, Inc. USA). The membrane was kept in blocking solution for 2 hrs [Tris-buffered saline containing Tween-20; HiMedia) (TBS-T) (0.1%) containing Non-fat milk 5%; Cat# M7409, Merck, India]. Membrane was gently washed with TBS-T (5 min/wash) and then incubated in any one of the following primary antibody for about 12-16 hrs at 4˚C. Anti-ITSN1mouse polyclonal antibody (Cat# SC-136242; Dilution: 1:100; Santa Cruz Biotechnology, Inc.); anti-HTT rabbit polyclonal antibody (Cat # ABP54729, Dilution: 1:2000; Abbkine); anti-SYT4 rabbit polyclonal antibody (Cat # ABP60576, Dilution: 1:500; Abbkine); rabbit polyclonal anti-BDNF (N-20) antibody (Cat # SC-546, Dilution: 1:200; Santa Cruz Biotechnology, Inc.); anti-RELN rabbit polyclonal antibody (Cat# ABP60127, Dilution: 1:1000; Abbkine); anti-ApoER2 rabbit polyclonal antibody (Cat # ABP55811, Dilution: 1:500; Abbkine); anti-VLDLR rabbit polyclonal antibody (Cat # ABP60892, Dilution: 1:500; Abbkine); rabbit polyclonal anti-SFK antibody (Cat # SC-8056; Dilution: 1:200; Santa Cruz Biotechnology, Inc.); anti-Dab1 rabbit polyclonal antibody (Cat # CST-3328; Dilution: 1:1000; Cell Signaling Technology, Inc.); anti-Phospho-Dab (Tyr 220) rabbit polyclonal antibody (Cat # 3327, Dilution: 1:1000; Cell Signaling Technology, Inc.); rabbit polyclonal anti-Akt3 antibody (Cat # PG-01303, Dilution: 1:2000; Puregene); rabbit polyclonal anti-Phospho-Akt (Thr 308) antibody (Cat # 9275L, Dilution: 1:1000; Cell Signaling Technology, Inc.); rabbit polyclonal anti-GluR1 antibody (Cat # ABP51437, Dilution: 1:500; Abbkine); anti-GluR2 rabbit polyclonal antibody (Cat # ABP514380, Dilution: 1:500; Abbkine); rabbit polyclonal anti-MEK-1 (C-18) antibody (Cat # SC-219, Dilution: 1:2000; Santa Cruz Biotechnology, Inc.); anti-ERK 1/2 (C-14) rabbit polyclonal antibody (Cat# SC-154, Dilution: 1:200; Santa Cruz Biotechnology, Inc.), as a control rabbit polyclonal anti-ß-actin antibody (Cat# SC-47778, Santa Cruz biotechnology, Inc.) was used. The membrane was washed in 1XTBS-T (3 X 5 min) and specific protein binded antibodies were detected by secondary

antibody alkaline phosphatase (ALP) conjugated either goat anti-rabbit IgG-ALP (GeNei[TM], Dilution: 1:5000; Cat# 621100180011730) or goat anti-mouse IgG-ALP (GeNei[TM], Dilution: 1:5000; Cat# 105215). The membrane was washed with 1XTBS-T (2x 5 min) and membrane bound ALP activity was determined with alkaline phosphatase substrate (AP Detection Reagent Kit, Merck). Molecular Imager was used to obtain the images and the specific band intensity was calculated, (ChemiDoc XRS;Image Lab-2, Bio-Rad Laboratories, Inc., USA) and data were dispayed as relative difference to control in fold. Representative uncropped western blot images were shown in S1 File.

## RNA isolation

Total RNA was isolated (Purezol, Bio-Rad Laboratories, Inc), and dissolved in Diethyl pyro-carbonate (DEPC;Cat#D5758; Sigma-Aldrich, India) water (50μl) and stored at -80˚C. Ran-dom/ oligo-dT primers (iScript[TM] cDNA synthesis kit; Bio-Rad Laboratories, Inc) was used to synthesis cDNA and stored at 4˚C.

## Quantitative Real Time (qRT) –PCR

Quantitative RT-PCR was performed (CFX-96, Bio-Rad Laboratories, Inc.) using SYBR[®] Green supermix (Bio-Rad laboratories, Inc.) and primers (100 pM) [BDNF-III F: 5′–TGAG ACTGCGCTCCACTCCC–3′, R:5′CGCCTTCATGCAACCGAAGTAT–3′; BDNF-IV F: 5′– CAGAGCAGCTGCCTTGAT GTTT–3′, R:5′–CGCCTTCATGCAACCGAAGTAT– 3′ and ß-actin F:5′–GCCAGAGCAGT AATCTCCTTCT–3′, R:5′–AGTGTGACGTTGACATCCGTA– 3′] [46]. PCR reaction set initially at 92˚C for 30 s, 92˚C for 5 s; annealing (BDNF-III: 63.6˚C; BDNF-IV: 56˚C; ß-actin: 55˚C) for 5 s and 72˚C for 5 s as extension. Amplification was con-firmed by analyzing dissociation and melt curve. Expression level was calculated in relative fold change to control. Triplicate reactions were performed and confirmed with native poly-acrylamide gel (10%).

## Immunohistochemistry (IHC)

Whole brain was fixed (50% formaldehyde;Cat#1.94989.5021; Merck India) for 24 hrs then transferred to increasing graded concentration of isopropanol;Cat#1.94524.0521; Merck India) (70%-100%). Samples were treated with xylene (Cat# 60868505001730; Merck India) 45 min, then processed in wax (Cat# 61782305001730; Merck India) for 10-12 hrs and the blocks were prepared. Cross sections (5μ) were prepared using the microtome (Weswox optik model – 1090A 15236) and fixed on slides. The slides were kept in warmer condition at 80˚C (5 min), transferred to xylene (15 min) and then to isopropanol (15 min). After that slides were washed in saline (PBS: 5 min) and then treated with peroxidase. Subsequently, washed with trypsin (Cat# RM713; HiMedia, India) (0.1%) and $CaCl_2$ (Cat# RM710; HiMe-dia, India) (0.1%) for 20 min. The slides were incubated with rabbit polyclonal anti-BDNF antibody (Cat# SC- 20981, Dilution: 1:50 Santa Cruz Biotechnology, Inc.) for 16 hrs and then goat anti-rabbit-horse radish peroxidase (HRP) (Cat# SC- 2030, Dilution: 1:5000; Santa Cruz Biotechnology, Inc.) for 12 hrs. Sections were stained using peroxidase develop-ment kit (Vector Laboratories, Inc. USA) and counterstained with hematoxylin (HiMedia, India) for 10 min and mounted with DPX. Microscopic Images of BDNF expressing neu-rons were obtained (Eclipse Ci-plus, Nikon instruments Inc., Japan) using DS-Fi3 camera and NIS Element - D Software. Number of BDNF positive neurons in hippocampus dorsal and ventral dentate gyrus was counted from selected sections (6 section/sample; n=3 from each group) and represented in percentage [47].

## Gas Chromatography/Mass Spectrometry (GC-MS) analysis

**Sample preparation.**   Tissue homogenization ($25\pm5$mg from hippocampus) was done using of chloroform/methanol/water solvent mixture (500 µL) in the ratio of 2:5:2, then centrifugation was done at 13,000 rpm at 4˚C for 15 min. The supernatant was collected then lyophilized at -70˚C (1hr) and vacuum pressure (5 *Pa*). The samples were mixed with methoxylamine hydrochloride (Cat# 226904; Sigma-Aldrich, India) in anhydrous pyridine (25 µl of 15 mg/ml), then ultrasonicated for 15 min at 37˚C. Subsequently incubated (70˚C) in water bath for 1 hr and mixed with (25µl) of bis-(trimethylsilyl) trifluoroacetamide (BSTFA) reagent in 1% trimethylchlorosilane (TMCS) followed by incubation at 70˚C (1hr). Then centrifugation was done at 13,000 rpm for (15 min) and the supernatant was collected.

The homogenization of feces samples ($40 \pm 5$ mg) were done using of chloroform/methanol/water solvent (800 µL) mixture in the ratio of 2:5:2 and centrifugation was done at 12,000 rpm at 4˚C for 15 min. The supernatant was collected carefully and mixed with (5µl) of pentadecane (Cat# 76610; Sigma-Aldrich, India) and of pyridine (5 µl) (Cat#270970; Sigma-Aldrich, India). The samples were then pre-chilled at $-80$˚C (1 hr), lyophilized ($-70$˚C, 1hr) at vacuum pressure (5 *Pa*). The lyophilized samples were mixed with methoxylamine hydrochloride (Cat# 226904; Sigma-Aldrich, India) in anhydrous pyridine (30 µl of 20 mg/ml), incubated (37˚C for 90 min) after that treated with 30 µl of bis-(trimethylsilyl) trifluoroacetamide (BSTFA) reagent (1%) with trimethylchlorosilane (TMCS) (Cat# 15238; Sigma-Aldrich, India) followed by incubated (70˚C for 1hr) in and then dissolved in methanol (500 µl).

**GC-MS analysis.**   The SH-Rxi-5 Sil MS column (30 m × 0.25 mm × 0.25 µm) with Helium gas as a carrier in inlet purge flow (3.0 ml/min) and column gas flow (1 ml/min) with GC-MS (GCMS-QP2020, Shimadzu Pvt. Ltd.,). Sample (1µl) was injected using the split mode, with an oven temperature (initially at 70˚C for 2 min; 160˚C with 6˚C/min; then 240˚C with 10˚C/min; and finally 300˚C (20˚C/min) for 6 min. The parameters such as temperatures of injector, transfer line, and electron impact on ion source were set to 250, 290, and 230˚C respectively. The GC/MS data was processed (GC-MS solutions version 4.45, Lab solutions) metabolites were identified process with http://www.metaboanalyst.ca for generating heat map.

**Statistical analysis.**   GraphPad Prism (ver 7.0) was used for graphical representation (**X̄±SD**). For all analysis data were primarily tested for normality and homogeneity, which is pre-request for the following analysis. Data were analysed with Two-way analysis of variance (ANOVA; Sigma Stat, ver 11.0) (behaviour, immunohistochemistry) and One-way ANOVA (molecular data), followed by pairwise t-test with Berferroni *post hoc* test.

## Ethics statement

Institutional Animal Ethical Committee (Institutional Animal Care & Use Committee: Ref. No: BDU/IAEC/RE01/2021) reviewed and approved the animal trapping and all experimental procedure following the regulations of CPCSEA, Govt. of India. The study animal *Mus booduga* categorized as least concern by International Union for Conservation of Nature [48]. The experiment designed in this study intended to minimize animal number and their sufferings.

## Results

### Environmental enrichment reverses the Social Isolation (SI) induced memory impairment

Hole-board test (HBT) was used to examine whether EE reverse the SI induced reference and working memory impairment. We observed significant interaction between housing condition × days ($F_{4,37} = 16.00$, $P<0.001$) in their learning abilities, further analysis showed

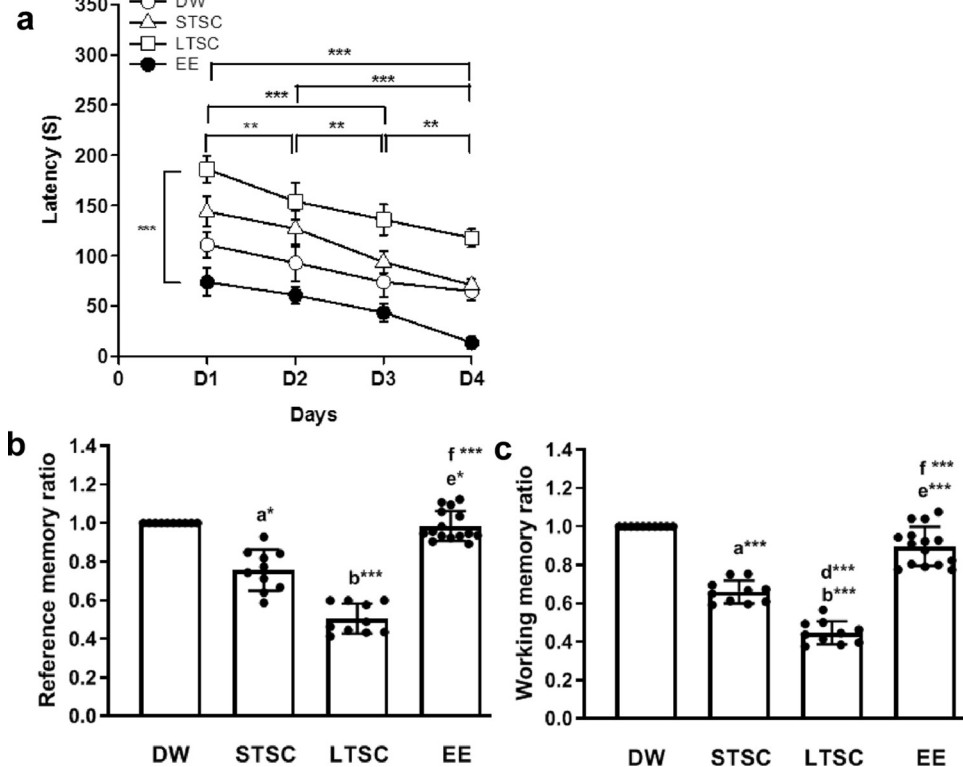

**Fig 1. Environmental enrichment improves the social isolation induced learning impairment.** Field mice *Mus booduga* behavioural response in the hole-board test showed that (**a**) latency to the baited hole was decreased significantly when they are learning from day-1 to 4 during the learning phase, thus, indicating that these mice were able to learn. However, there were significant effects of housing condition on latency to baited hole. Ratio of (**b**) Reference memory (**c**) Working memory is influenced by the housing condition, EE housing condition significantly reversed the memory impairment induced by social isolation in STSC housing condition. Data presented as mean ± SEM [DW-Direct Wild (n = 10); STSC- Short Term at Standard Condition (n = 10); LTSC- Long Term at Standard Condition (n = 10); EE- Environmental Enrichment (n = 15)], Two way ANOVA and One way ANOVA with *post hoc* comparisons [a: DW vs STSC; b: DW vs LTSC; c: DW vs EE; d: STSC vs LTSC; e: STSC vs EE; f: LTSC vs EE] and * indicates level of significant (*p<0.05;**p<0.01;***p<0.001).

the significant difference in their learning abilities between the animals housed in different conditions following SI ($F_{4,37}$ = 143.063, $P<0.001$), between the testing days ($F_{4,37}$ = 57.8, $P<0.001$) (Fig 1A). Notably, we have observed significant difference in reference memory between the animals housed in different conditions following SI ($F_{3,44}$ = 189.377; $P<0.001$). *Post hoc* analysis indicates greater effect of SI induced impairment in reference memory was observed in the LTSC mice, and they made more number of errors compared to DW ($P<0.001$), EE ($P<0.001$) (Fig 1B). STSC mice made more error than DW ($P<0.05$). However, no difference between DW ($P = 0.89$) and STSC ($P<0.05$) with EE and between STSC and LTSC mice ($P=0.574$). Similarly, we have observed that SI influences the working memory, thus, significant difference was observed in working memory between the mice housed in different condition ($F_{3,44}$ = 159.093, $P<0.001$). *Post hoc* analysis showed that LTSC mice had poorer working memory than DW ($P<0.001$), STSC ($P<0.001$), EE mice ($P<0.001$). STSC mice experienced stress, hence they made more error than DW ($P<0.001$) and EE ($P<0.001$). Significant difference not detected between DW and EE ($P = 0.055$) mice (Fig 1C). Collectively, these results suggest that EE reversed SI induced impairment in reference and working memory. Therefore, we examined genes associated with the synaptic plasticity.

## EE reverses the SI induced alteration in expression of intersectin-1 and its associated signaling molecules

Intersection 1 (ITSN 1) is a component of Reelin signalling pathway that has been implicated in hippocampal dependent synaptic plasticity. Therefore, at first we examined the ITSN1 and associated downstream signalling molecules. We found significant difference in the expression of ITSN1between the mice housed in different condition ($F_{3,23}$ = 167.804,$P<0.001$). *Post-hoc* test showed that ITSN expression was significantly elevated in LTSC, STSC mice than DW ($P<0.001$) and EE mice ($P<0.001$). ITSN level was significantly different between STSC and LTSC mice ($P<0.001$), but not between DW and EE mice ($P$ =0.582) (Fig 2A and 2B; S2 Fig in S1 File). ITSN has been known to co-localize with Htt, estimated level of Htt was significantly different between the mice housed in different conditions ($F_{3,23}$ = 17.987, $P<0.05$), estimated Htt level was significantly higher in LTSC mice than DW ($P<0.001$) and EE mice ($P<0.001$). However, significant difference was not detected when comparing LTSC vs STSC ($P$ = 0.575), DW vs STSC ($P$ = 1.000), DW vs EE ($P$ = 1.000) and STSC vs EE ($P$ = 1.000) (Fig 2A and 2C; S2 Fig in S1 File). Further, analysis showed that the expression level of SYT-4 was significantly different between housing condition ($F_{3,23}$ = 20.012, $P<0.001$). Significantly higher level of SYT-4 was observed in LTSC mice than DW ($P<0.001$), STSC ($P$ = 0.428) and EE mice ($P<0.001$). Whereas, there was no significant difference between STSC and DW mice ($P<0.001$), and EE mice ($P<0.001$) (Fig 2A and 2D; S2 Fig in S1 File). When the BDNF expression was estimated significant difference observed between them ($F_{3,23}$ = 68.235, $P<0.001$). Detected BDNF level in LTSC mice significantly lower than DW ($P<0.001$), STSC ($P<0.01$) and EE mice ($P<0.001$). Whereas, the level of BDNF in DW mice was not significantly different from EE mice ($P$ = 1.000), (Fig 2A and, 2E; S2 Fig in S1 File).

## EE reverses the SI induced alteration in BDNF variant transcription

Activity-dependent BDNF transcript expression can provides context-specific information associated with BDNF level and synaptic plasticity. To understand how Bdnf mRNA alters the level of BDNF protein, we examined the Bdnf transcripts containing exons III and IV from mice housed in different condition. The level of transcript III was significantly altered by SI, thus significant difference was observed between housing conditions ($F_{3,23}$ = 21767.634, $P<0.001$). *Post hoc* test showed that expression was significantly higher in LTSC mice than DW ($P<0.001$), STSC ($P<0.001$) and EE mice ($P<0.001$). In comparison, Bdnf III transcript significantly higher in EE mice than DW ($P<0.001$), STSC ($P<0.001$) mice. Significant difference not observed between STSC and DW mice ($P<0.001$) (Fig 3A and 3b, S3 Fig in S1 File). The level of Bdnf transcript VI was significantly altered by SI, therefore significant difference detected between housing condition ($F_{3,23}$ = 303.730, $P<0.001$), indeed the expression pattern was contrast to transcript III. Bdnf transcript VI level significantly suppressed by SI,hence, observed level in LTSC lower than DW ($P<0.001$) and EE mice ($P<0.001$). Interestingly, significant difference between DW, and STSC ($P<0.001$), but not detected EE versus DW mice ($P<0.01$) (Fig 3C and 3D, S3 Fig in S1 File). Our observation suggests that SI differently regulate the Bdnf transcript III and IV, whereas EE significantly reversed the SI induced effect.

## EE reverses the SI induced inhibition of BDNF expression in hippocampus

BDNF expression in hippocampus has been implicated in both inhibitory, excitatory synaptic transmission and activity-dependent synaptic plasticity. Especially, balancing act of dorso-ventral axis of dentate gyrus (DG) regulates innate anxiety and learning [49]. Therefore, we examined immunoreactive pattern of BDNF in DG. We have observed higher level of BDNF

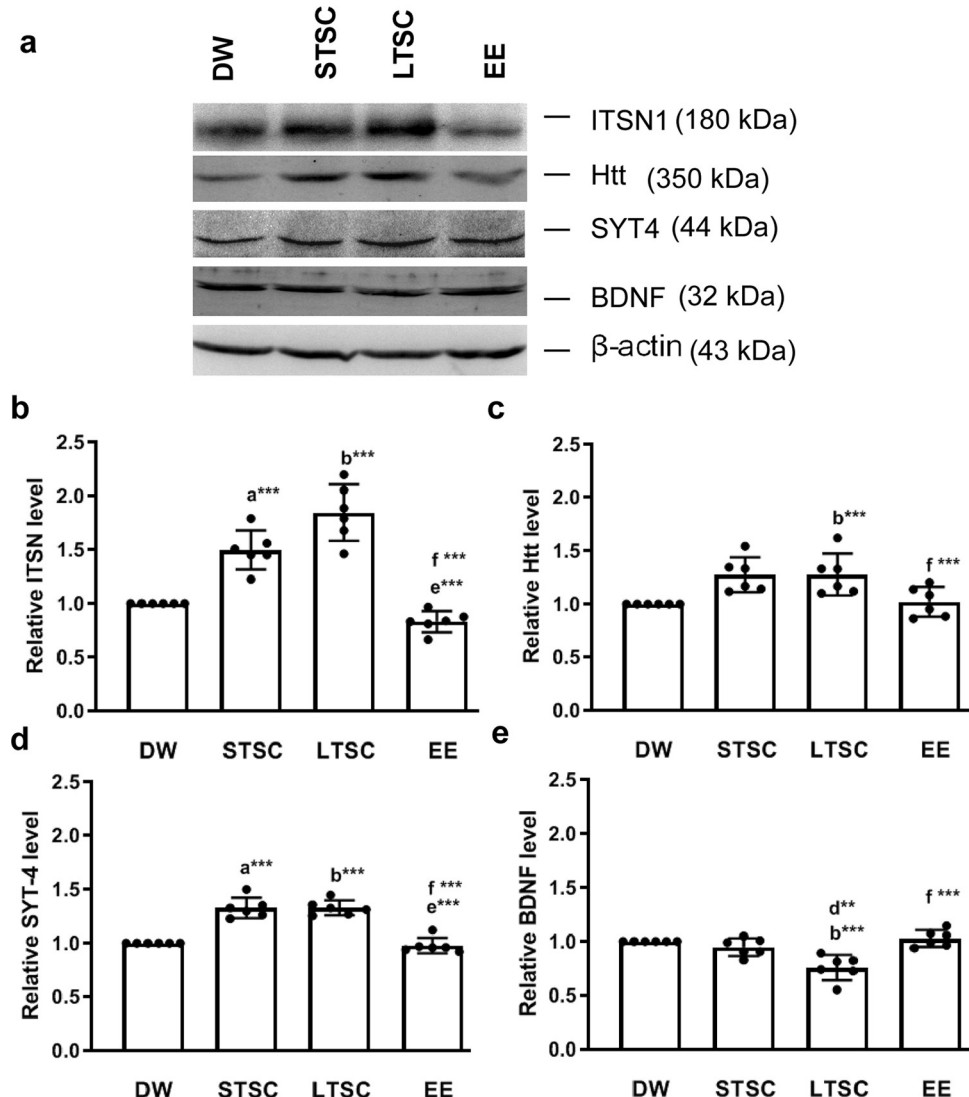

**Fig 2. Environmental enrichment reverses social isolation induced alteration in expression of ITSN1 and associated signalling molecules. (a)** Representative western blots showing the expression level of ITSN1 (180 kDa), Htt (350kDa), SYT4 (44 kDa) and BDNF (32 kDa). Estimated values showed that EE significantly reversed the social isolation induced alteration in the expression of **(b)** ITSN1, **(c)** Htt, **(d)** SYT4 and **(e)** BDNF. Data presented as mean ± SEM [DW-Direct Wild (n = 6); STSC- Short Term at Standard Condition (n = 6); LTSC- Long Term at Standard Condition (n = 6); EE- Environmental Enrichment (n = 6)], One way ANOVA with *post hoc* comparisons [a: DW vs STSC; b: DW vs LTSC; c: DW vs EE; d: STSC vs LTSC; e: STSC vs EE; f: LTSC vs EE] and * indicates level of significant (**p<0.01;***p<0.001).

immunoreactives in DW and EE mice dentate gyrus region of hippocampus, which was much higher compared to STSC and LTSC mice (Fig 4). We have observed significant interaction in the expression level between housing condition × dorso/ventral ($F_{4,47}$ = 87.61, $P<0.001$), further analysis showed the significant difference in between the housing conditions ($F_{4,47}$ =11.03, $P<0.01$) and between dorsal and ventral DG ($F_{4,47}$ =7.2, $P<0.01$). Further analysis showed that number of BDNF positive neurons is lower in STSC ($P<0.001$), LTSC ($P<0.001$) and DW mice ($P<0.001$) than EE mice. In comparison, intensity was significantly lower in STSC ($P<0.001$), LTSC ($P<0.001$) than EE. However, estimated level was not significantly different between STSC and LTSC mice ($P = 0.465$). In addition, we detected significant

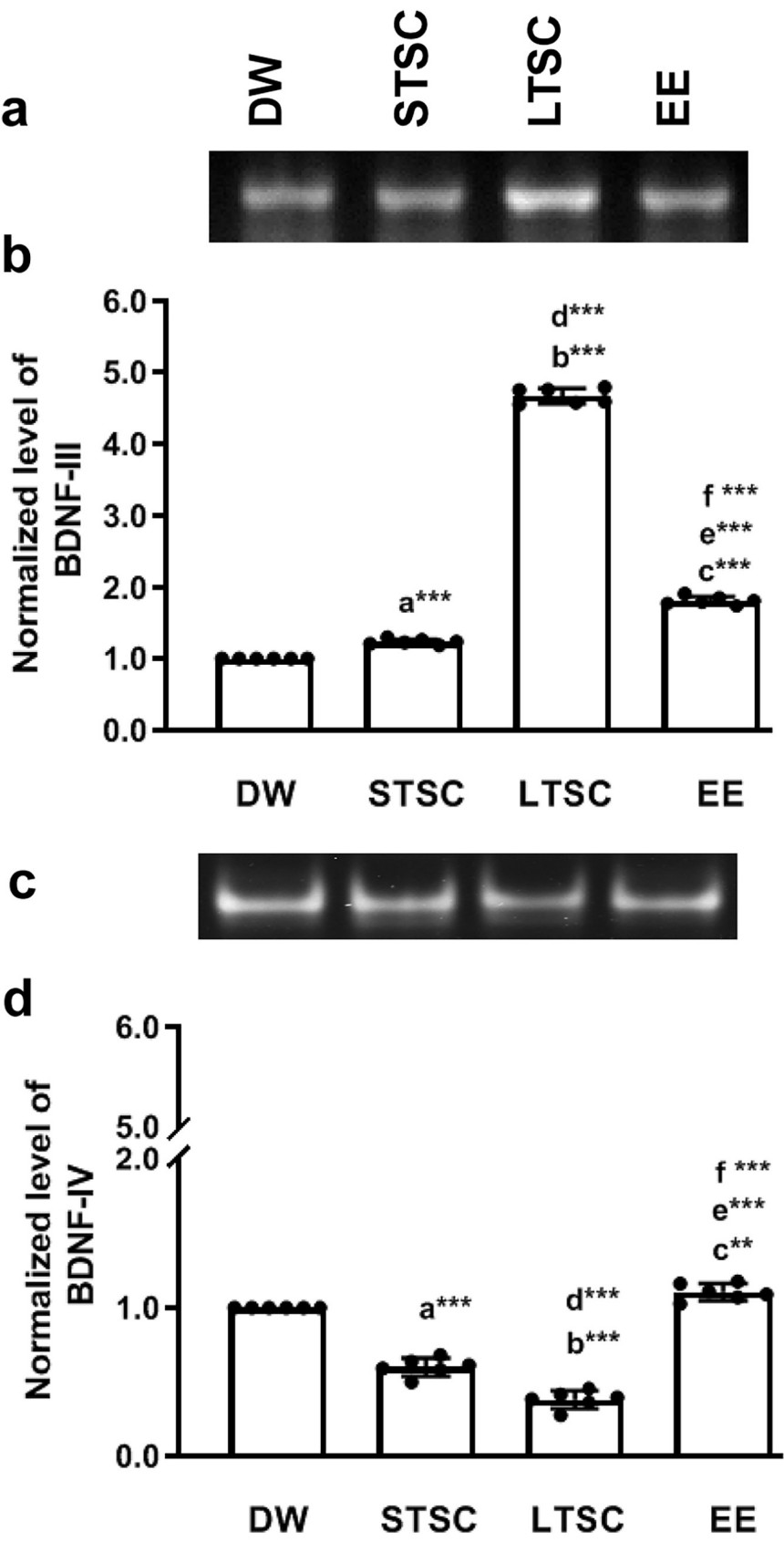

**Fig 3. Social isolation induced alteration in expression of Bdnf variant reversed by environmental enrichment.** Bdnf variants exon-III **(a)** and **(c)** exon -IV expression pattern in mice represents different housing condition. Real-time qPCR showing the expression of **(b)** Bdnf exon (III) level was increased, but **(d)** exon (IV) level was decreased by SI stress in STSC and LTSC housing condition, EE reversed the SI effect. Data presented as mean ± SEM [DW-Direct Wild (n = 6); STSC- Short Term at Standard Condition (n = 6); LTSC- Long Term at Standard Condition (n = 6); EE-Environmental Enrichment (n = 6)], One way ANOVA with *post hoc* comparisons [a: DW vs STSC; b: DW vs LTSC; c: DW vs EE; d: STSC vs LTSC; e: STSC vs EE; f: LTSC vs EE] and * indicates level of significant (**p<0.01;***p<0.001).

difference between dorsal and ventral dentate gyrus of STSC ($P<0.001$) and EE mice ($P<0.001$) was significant, but not in DW ($P = 1.0$) and LTSC mice ($P<0.001$) (Fig 4D). Observed expression pattern of BDNF neurons in hippocampus suggest that sensory stimulus in EE/ SI could influence hippocampus function.

## EE reverses the SI induced alteration in expression of reelin and associated molecule

BDNF has been known to interact with reelin, an extracellular glycoprotein. Reelin through its receptor controlling cell-cell interaction and modulate synaptic plasticity. We found that expression of Reelin was significantly altered by SI and subsequent different housing condition ($F_{3,23} = 191.751$, $P<0.001$). Further examination showed that level Reelin was significantly lower in LTSC mice than DW ($P< 0.001$), STSC ($P< 0.001$) and EE mice ($P< 0.001$). Reelin level was significantly lower in STSC than DW ($P<0.001$), EE mice ($P<0.001$) (Fig 5A and 5B; S4 Fig in S1 File). Subsequently, we examined the level of Reelin receptors ApoER2 and VLDLR. The analysis showed that SI and housing condition altered the ApoER2 expression ($F_{3,23} = 66.097$, $P<0.001$). *Post hoc* test revealed that the level of ApoER2 was significantly lower in LTSC mice than DW ($P<0.001$) and EE mice ($P<0.001$). Similarly, level of ApoER2 in STSC mice was significantly lower in DW ($P<0.001$) and EE mice ($P<0.01$), but significant difference not detected between the mice housed in LTSC and STSC ($P = 0.705$), EE and DW ($P = 0.503$) (Fig 5A and 5C; S4 Fig in S1 File).

Expression of VLDLR was significantly altered by different housing condition ($F_{3,23} = 489.710$, $P<0.001$). *Post hoc* test showed that expression of VLDLR was significantly lower in the mice housed in LTSC than DW ($P<0.001$) and EE mice ($P<0.001$). Likewise, level of VLDLR was significantly lower in STSC mice than DW ($P<0.001$) and EE mice ($P<0.001$). Estimated difference was not significantly different between LTSC and STSC mice ($P = 1.000$), DW and EE mice ($P = 1.000$) (Fig 5A and 5D; S4 Fig in S1 File). We then investigated the level of Src-family kinases (SFKs), the level was significantly altered by housing conditions ($F_{3,23} = 405.996$, $P<0.001$), *post hoc* test depicts that the expression of SFKs was significantly lower in STSC ($P<0.001$), LTSC ($P<0.001$), and EE mice ($P<0.001$) than DW mice. In comparison, significantly lower expression level was detected in STSC ($P<0.01$) and LTSC mice ($P<0.001$) than EE mice (Fig 5A and 5E; S4 Fig in S4 File). Obtained results suggest that SI effectively alter the expression of BDNF, Reelin, ApoER2, VLDR and SFKs, subsequent housing at EE reversed the effect but not the standard housing condition.

## EE reverses the SI induced alteration in phosphorylation of Dab1 and Akt

Further, activated Reelin and its receptor complex induce the phosphorylation of Dab1 and activate Dab1-dependent downstream signaling events including Akt/P13 kinases. Subsequently, we have examined phosphorylation of Dab1 and Akt, and found that housing condition significantly alters the expression and phosphorylation of Dab-1 ($F_{3,23} = 227.215$, $P<0.001$), the estimated lower level of relative p-Dab in LTSC mice than DW ($P<0.001$) and EE mice ($P<0.001$). The estimated relative p-Dab level in STSC ($P<0.001$) and EE mice

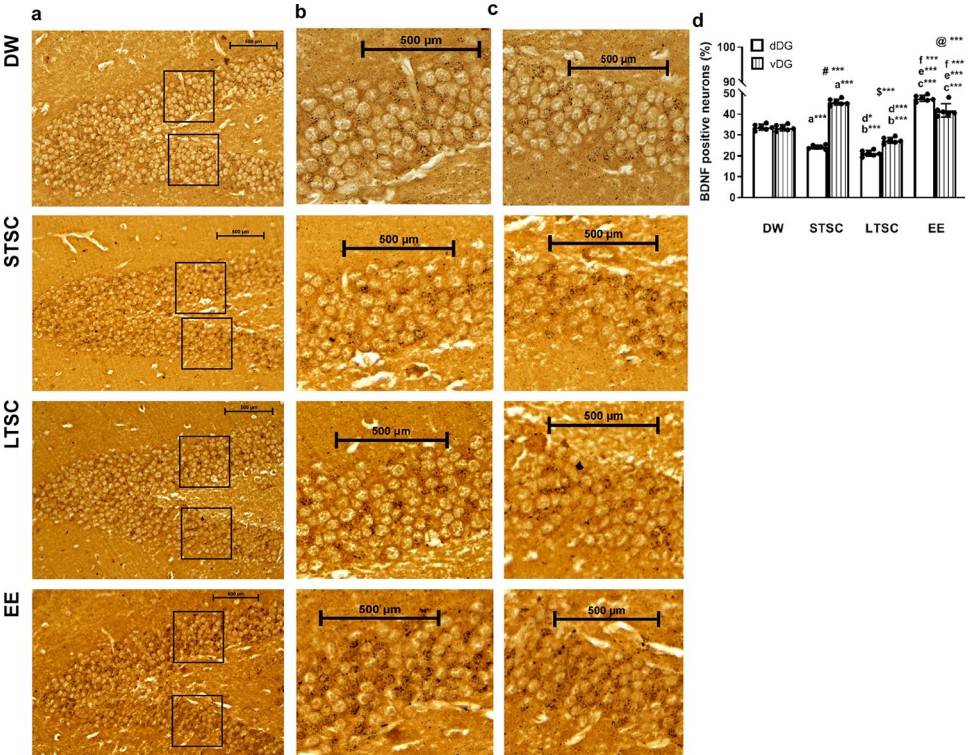

**Fig 4. Environmental enrichment reverses the social isolation induced changes in the expression of BDNF in hippocampus.** Light micrograph (40X) of hippocampus region showing the expression pattern of BDNF positive neurons (**a**), higher magnification (40X) of selected (**b**) dorsal, (**c**) ventral region of the box in "a" and (**d**) Optical density of BDNF immunoreactivity in granular layer of dorsal dentate gyrus (dDG) and ventral dentate gyrus (vDG). Scale bars: 500μm (a-c). Significant difference between dDG and vDG showing # for DW and $ for EE mice. Data presented as mean ± SEM [DW-Direct Wild (n = 4); STSC- Short Term at Standard Condition (n = 4); LTSC- Long Term at Standard Condition (n = 4); EE- Environmental Enrichment (n = 4)], One way ANOVA with *post hoc* comparisons [a: DW vs STSC; b: DW vs LTSC; c: DW vs EE; d: STSC vs LTSC; e: STSC vs EE; f: LTSC vs EE] and * indicates level of significant (*p<0.05;***p<0.001).

($P$<0.001) was lower than DW mice. Further, the level was significantly different between EE and DW mice ($P$<0.001), but there was no significant difference between LTSC and STSC mice ($P$ = 0.249) (Fig 6A and 6B; S5 Fig in S1 File). Similarly, the level of Akt phosphorylation was significantly altered by housing condition ($F_{3,23}$ = 42.390, $P$<0.01). The relative phosphorylation level of Akt in LTSC lower than DW ($P$<0.001), STSC ($P$<0.05), EE mice ($P$<0.001). But not in any other comparisons DW vs EE mice ($P$ = 1.000), DW vs STSC ($P$ = 1.000) and STSC vs EE mice ($P$ = 0.291) (Fig 6A and 6C; S5 Fig in S1 File). Observed phosphorylation of Dab1 and Akt suggest that SI effectively suppress the phosphorylation, which is further maintained by STSC, LTSC housing condition. Whereas EE reversed SI induced effect and restored the level of Dab1 and Akt.

## EE reverses the SI induced alteration in expression of AMPAR subunits GluR1 and GluR2

AMPA glutamate receptors are key mediators of excitatory transmission fine tune the synaptic plasticity. Hence, we examined the level of GluR1 and GluR2 in mice housed under different condition (Fig 7A; S6 Fig in S1 File). The analysis shows that the housing condition significantly altered the expression of GluR1 ($F_{3,23}$ = 549.608, $P$<0.001) (Fig 7B) and GluR2 ($F_{3,23}$ =

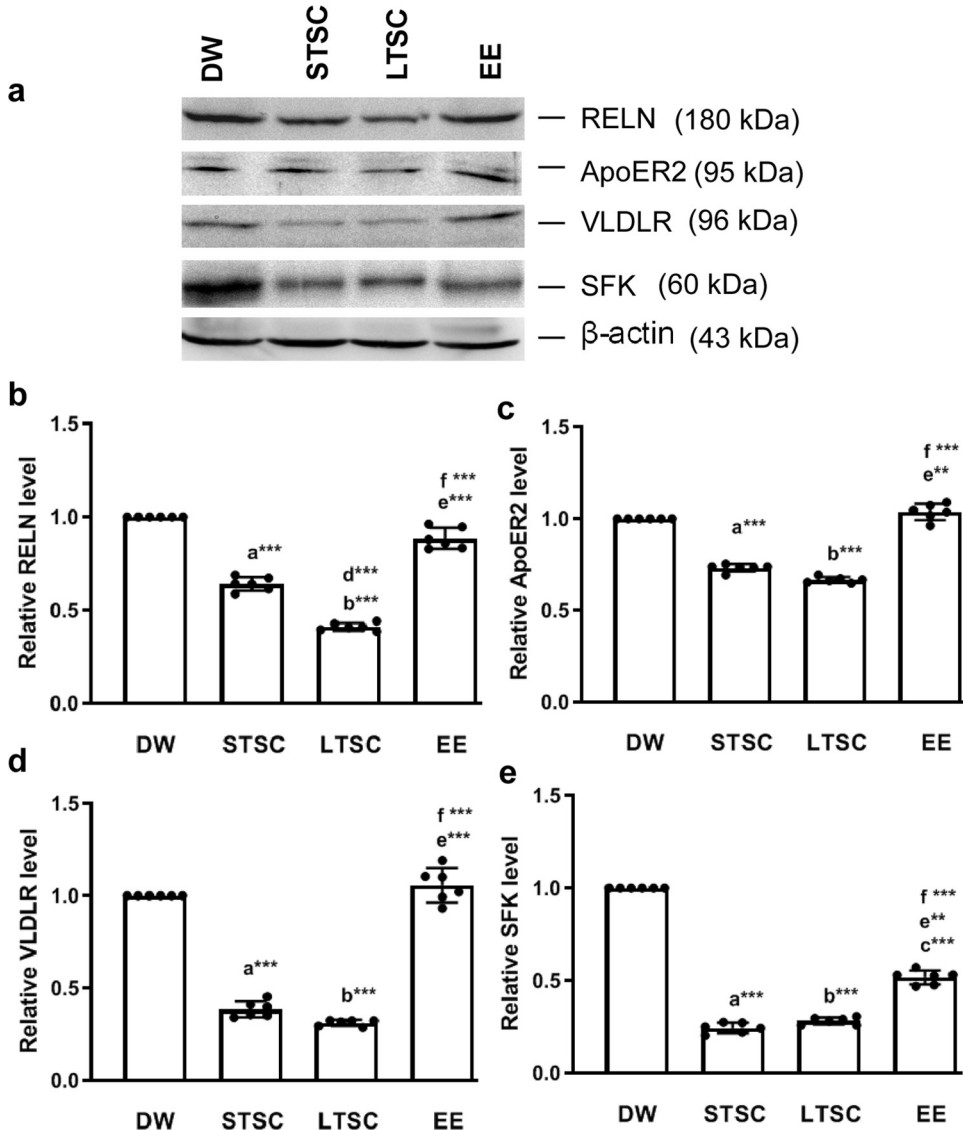

**Fig 5. Environmental enrichment reverses the social isolation induced alteration in expression of Reelin, its receptors and SFK. (a)** Representative western blot showing that expression level of Reelin (180 kDa), APoER2 (95 kDa), VLDLR (96kDa) and SFK (60 kDa). EE housing condition reversed the SI induced stress reduced the expression of **(b)** Reelin, **(c)** APoER2, **(d)** VLDLR and **(e)** SFK. Data presented as mean ± SEM [DW-Direct Wild (n = 4); STSC-Short Term at Standard Condition (n = 4); LTSC- Long Term at Standard Condition (n = 4); EE- Environmental Enrichment (n = 4)], One way ANOVA with *post hoc* comparisons [a: DW vs STSC; b: DW vs LTSC; c: DW vs EE; d: STSC vs LTSC; e: STSC vs EE; f: LTSC vs EE] and * indicates level of significant (**p<0.01;***p<0.001).

838.470, *P<0.001*) (Fig 7C). Expression of GluR1 and GluR2 was similar, the level in LTSC mice was significantly higher than DW (*P<0.001*) / EE mice (*P<0.001*), STSC mice (*P<0.05*). In comparison, the expression level between DW and EE mice was not significantly different (GluR1 *P* = 0.282; GluR2 *P* = 1.000), however, in STSC mice only the GluR1 expression was higher than DW (*P<0.001*) and EE mice (*P<0.001*), but not in case of GluR2 STSC vs DW mice (*P* = 0.376) and STSC vs EE mice (*P* = 0.847). This analysis showed that SI influences the expression of GluR1 and R2, but subsequently housing at EE reversed the effect.

In parallel, through Dab1 phosphorylation Reelin activates MAPK and ERK1/2 signaling pathway for the transcriptional activation of candidate genes. Subsequently, we found that

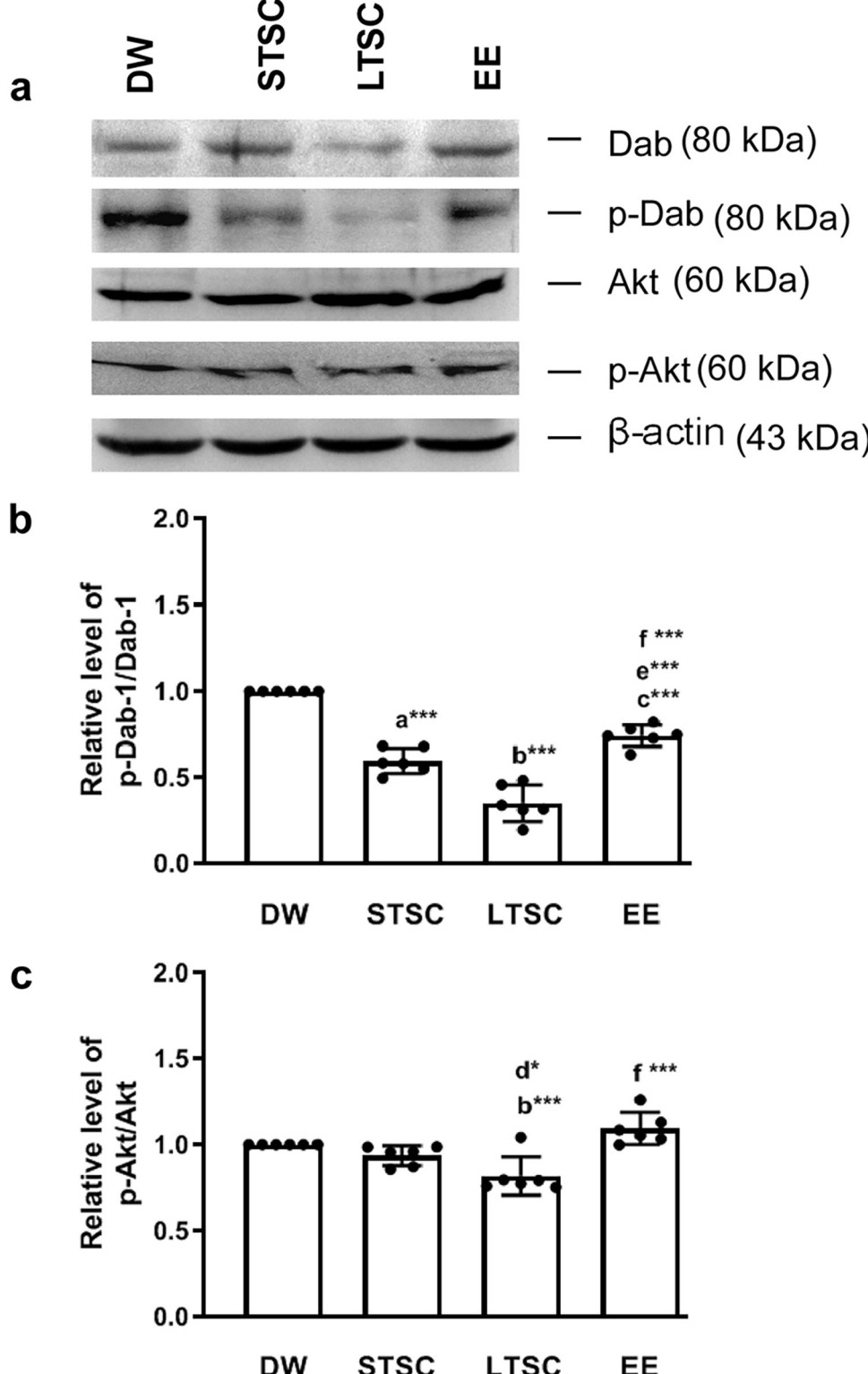

**Fig 6. Environmental enrichment reverses the social isolation induced changes in phosphorylation of Dab-1 and Akt levels. (a)**Western blot images shows expression pattern of Dab-1 and Akt in mice experienced different housing condition. Estimated values show that environmental enrichment reverse social isolation changes in the Level of pDab-1/Dab1 **(b)** and pAkt/Akt (c). Data presented as mean ± SEM [DW-Direct Wild (n = 4); STSC- Short Term at Standard Condition (n = 4); LTSC- Long Term at Standard Condition (n = 4); EE- Environmental Enrichment

(n = 4)], One way ANOVA with *post hoc* comparisons [a: DW vs STSC; b: DW vs LTSC; c: DW vs EE; d: STSC vs LTSC; e: STSC vs EE; f: LTSC vs EE] and * indicates level of significant (*p<0.05;***p<0.001).

expression of MAPK was significantly altered by housing condition ($F_{3,23}$ = 80.101, $P<0.001$), *post hoc* test showed that the expression of MAPK was significantly was lower in LTSC mice compared to DW ($P<0.05$), STSC ($P<0.05$) and EE mice ($P<0.05$). However the detected variation in the MAPK level was not significantly different in any comparison between DW, STSC and EE mice (DW vs STSC: $P$ = 1.000, DW vs EE: $P$ = 0.799, STSC vs EE: $P$ = 0.839) (Fig 7A and 7D; S6 Fig in S1 File). Further, the estimated level of ERK1/2 in mice under different housing conditions was significantly different ($F_{3,23}$ = 110.910, $P<0.001$). We detected lower level of ERK expression in the mice housed at LTSC compared to the DW ($P<0.05$), STSC ($P<0.05$) and EE mice ($P<0.05$), whereas the estimated level was higher in EE than STSC mice ($P<0.05$) However, the estimated variation in the expression of ERK1/2 was not significantly different between DW, STSC and EE mice (DW vs STSC: $P$ = 0.105, DW vs EE: $P$ = 0.775) (Fig 7A and 7D; S6 Fig in S1 File). Our analysis suggests that EE reverse the SI induced changes in MAPK and ERK1/2 expression.

## EE reverses the SI induced alteration in metabolites in hippocampus and feces

Bidirectional communication between central nervous system and gut facilitated through active metabolites. Therefore, we examined the metabolites from hippocampus and feces. We found that the level of metabolite was altered by housing conditions. Based on the earlier reported function, we have grouped them as anxiety and anxiolytic metabolites. Heatmap analysis segregates based on the estimated baseline values, which support their physiological role (Fig 8A and 8B). Two different metabolites benzenedicarboxylic acid and 2-pregnane are known to be involved in the pathways that contribute the anxiety or anxiolytic-like behaviour. The estimated level of Benzenedicarboxylic acid was significantly different in hippocampus ($F_{3,23}$ = 838.8, $P<0.001$) and feces ($F_{3,23}$ = 692.1, $P<0.001$) from the mice housed in different conditions. The *post hoc* analysis suggests that lower level in STSC ($P<0.001$), LTSC ($P<0.001$) and DW mice ($P<0.001$) compared to EE mice. Whereas the detected level was significantly lower in LTSC than STSC ($P<0.001$) and DW mice ($P<0.001$), but no difference between STSC and DW mice ($P$ = 1.000). In hippocampus, metabolite level was significantly higher in DW than EE ($P<0.001$), the level was not detectable in STSC and LTSC mice (Fig 8C). The metabolite 2-Pregnane in both feces ($F_{3,23}$ = 64.637, $P<0.001$) and hippocampus ($F_{3,23}$ = 1.261, $P<0.001$) was significantly altered by different housing condition. The estimated level in LTSC was significantly different from STSC in feces ($P<0.001$) but not in brain ($P$ = 0.288). Notably, metabolite was not tracable in DW and EE mice samples (Fig 8D).

## Discussion

Social isolation (SI) and environmental enrichment (EE) are housing conditions frequently used to examine psychiatric endophenotypes, recapitulating the pathology and underlying mechanism [49]. In this study, we examined whether EE reverse the SI induced memory impairment and genes associated with synaptic plasticity in Indian field mice *Mus booduga*. We found that mice representing different housing conditions showed that they learned from day 1 to 4. Thus, their latency was reduced towards the baited hole in the HBT. However, compared to any other group the latency of the EE mice was significantly lower, and their reference and working memory ratio was higher. Supporting to our observations, earlier studies in

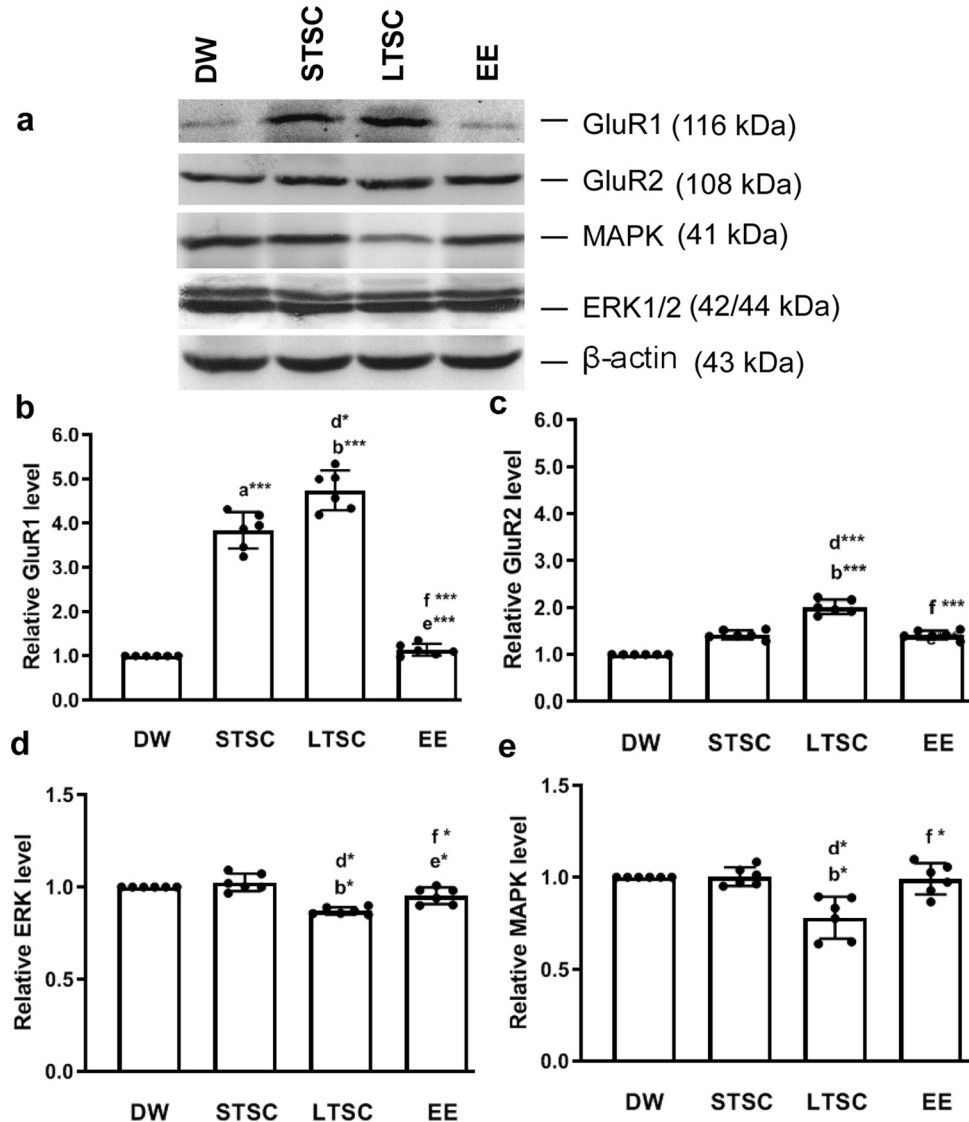

**Fig 7. Environmental enrichment reverses the social isolation induced changes in the expression of GluR1, GluR2, MAPK and ERK-1/2. (a)** Representative western blot images shows that expression pattern of GluR1 (116 kDa), GluR2 (108k Da), MAPK (41 kDa) and ERK-1/2 (42/44kDa). Estimated values shows that environmental enrichment significantly reveres the social isolation induced changes in the expression of **(b)** Glu R1, (c) GluR2, (d) MAPK and (e) ERK-1/2. Data presented as mean ± SEM [DW-Direct Wild (n = 4); STSC- Short Term at Standard Condition (n = 4); LTSC- Long Term at Standard Condition (n = 4); EE- Environmental Enrichment (n = 4)], One way ANOVA with *post hoc* comparisons [a: DW vs STSC; b: DW vs LTSC; c: DW vs EE; d: STSC vs LTSC; e: STSC vs EE; f: LTSC vs EE] and * indicates level of significant (*p<0.05;**p<0.01;***p<0.001).

laboratory animals demonstrated that EE resilience the SI induced impairment in working memory and reference memory [43, 50, 51], and spatial memory [29, 43]. Specifically, EE resilience SI induced anxiety-like behaviour [52] and improve working memory in field-caught rodents [32, 53]. Improved reference and working memory in EE mice possibly by the activation or improvement of long-term synaptic plasticity by enriched housing condition [5].

Hippocampus is known to synchronize with other regions of brain for processing and maintenance of memory [54]. Thus, we examined the signaling pathway associated with

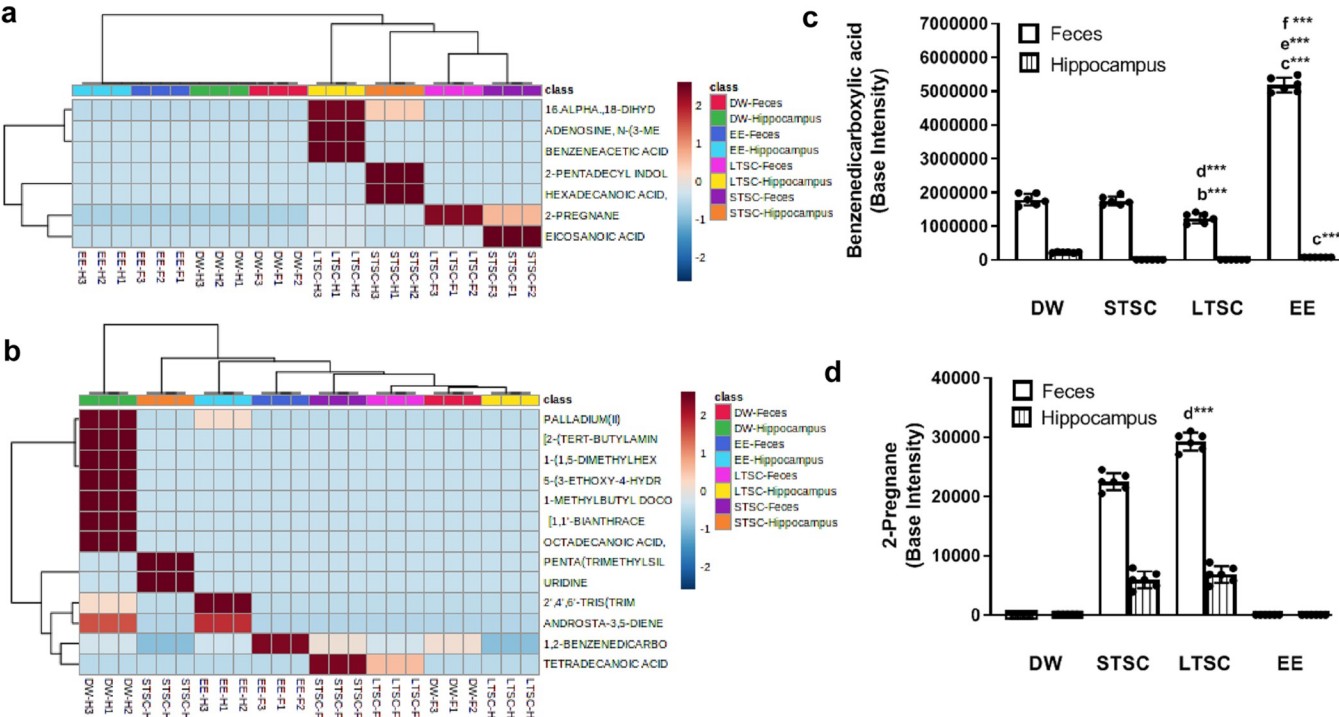

**Fig 8. Environmental enrichment reverses the social isolation induced changes in the gut and brain metabolities.** Heatmap analysis shows the changes in the metabolites in hippocampus tissue and feces of mice represent different housing condition. Cluster heatmap visualization shows that distinct segregation of metabolites that **(a)** induce anxiety-like behaviour and **(b)** anxiolytic behaviour identified from hippocampus tissue and feces of mice represent DW, STSC and LTSC and EE group. Colour key indicates the estimated level of metabolites: highest-red and lowest-blue. Social isolation induced changes in metabolites **(c)** Benzenedicarboxylic acid, and **(d)** 2- pregnane were reversed by environmental enrichment. Data presented as mean ± SEM [DW-Direct Wild (n = 6); STSC-Short Term at Standard Condition (n = 6); LTSC- Long Term at Standard Condition (n = 6); EE- Environmental Enrichment (n = 6)], Two way ANOVA with *post hoc* comparisons are [a: DW vs STSC; b: DW vs LTSC; c: DW vs EE; d: STSC vs LTSC; e: STSC vs EE; f: LTSC vs EE] and * indicates level of significant (***$p < 0.001$).

activation of long-term potentiation (LTP) and synaptic plasticity in hippocampus region. ITSN1 has been known to regulate pre-synaptic neurotransmitter release, which depends on recycling of release-ready exocytosis of synaptic vesicles (SVs) within the readily releasable pool (RRP) [8, 55], synaptic plasticity [8, 56], and learning and memory [9]. The expression of ITSN was higher in LTSC mice than other experimental group animals, EE reversed and normalizes the level. In fact, previous studies have indicated that a greater or lesser level of ITSN expression leads to memory impairment [57], which has altered recycling of synaptic vesicle (exocytosis/ endocytosis). Indeed, overexpression of ITSN1 was associated with the genotype of the huntingtin's disease, which showed aggregation of the huntingtin (htt) protein and neuronal dysfunction [11]. Observed memory deficits in LTSC mice possibly associated with higher expression of Htt. Synaptic plasticity is tightly regulated by the level of BDNF, a balanced expression of wild type Htt has been known to control synaptic plasticity through BDNF [58]. Note to mention that, HTT mediate transport of BDNF along the axon with the interaction of synaptic vesicle contain soluble N-ethylmaleimide-sensitive factor attachment protein receptors (SNARE) proteins in synaptotagmin and synaptobrevin [59]. In which, SYT-IV regulates the spontaneous transmission of BDNF at pre and post-synaptically [14]. Notably, reduction of SYT-IV enhances the release of BDNF and LTP, whereas, overexpression of SYT-IV increases the proportion of "non-productive" fusion events and release less BDNF. Indeed, in both condition the normal LTP is affected and lead to memory impairment

[15]. The observed memory impairment in LTSC mice was possibly due to the up-regulated SYT-IV, which could be linked with reduced BDNF release.

On the other hand, differential regulation of the BDNF III and IV promoters by SI and following the housing condition and physical activity could lead to reduction in total BDNF [60]. Supporting to this, we found that the transcript level of BDNF III was elevated but IV was reduced in LTSC mice, but EE reversed the SI induced effect [60, 61]. Observed decline in BDNF IV mRNA and total BDNF protein following LTSC is comparable with earlier clinical and animal models of depression/ anxiety [62, 63]. Dentate gyrus (DG) granule cells are sensitive to external stimuli i.e. physical activity and chronic stress [64, 65]. Observed difference in the BDNF possibly in the dDG and vDG was possibly by the differential regulation of SI induced stress or physical activity in the EE [66, 67]. However, the differential regulation in dDG/vDG BDNF depends on the context [68]. BDNF, a neurotrophin highly expressed in the hippocampus, and regulate the expression of many genes, including Reelin [69]. In this line, the observed memory impairment in LTSC mice may be related to the observed lower level of Reelin expression. Stress sensitive epigenetic regulation of Reelin promoter may be the factor [70]. Further, we have found that the levels of ApoER and VLDLR–Reelin receptors are reduced by SI, followed by housing at LTSC, which is possibly influenced by lower level of Reelin. Notably, EE reversed SI effect and regulate the expression of Reelin, further it activates its receptors (ApoER, VLDLR) and enhances the synapse formation and LTP [71]. Reelin has been known to exert its functions through SFK-Dab1/p-Dab1 and promote dendritic compartment, dendritic spine formation and growth [72]. We found that the reduction of Dab1/p-Dab1 in LTSC mice can cause defects in dendritic spine formation and a deficit in learning and memory [73]. The level of Dab1/p-Dab restored in EE mice, possibly by directional regulation of environmental stimulus that release both synaptic strengthening and pruning [74]. Further, the reduced level of Dab1/p-Dab1 in LTSC mice possibly recruits the minimal amount of Akt to the complex [75]. Reduced level of phosphorylated Akt in LTSC mice has been reported in anxiety/depressive-like behavioural phenotype [76] and deficit in working memory [77] and enhanced level of phosphorylated Akt in EE mice, which coincide with stronger acquisition and working memory of EE mice [78].

In this signaling pathway, AMPA receptors are implicated. Observed elevated level of AMPAR subunits GluR1/2 in LTSC mice perhaps through alternation in the AMPAR compartmentalization or stimulated by the elevated glucocorticoid level [79]. In addition, mGluRs has been known to influence the ERK1/2 and MAPK signaling cascade possibly through Cap-dependent translational mechanism [80]. Supporting to this, we found that both ERK1/2 and MAPK levels were significantly lower in LTSC mice. Differences in the compartmentalization of mGluRs in the LTSC group can alter the level of ERK1/2 and MAPK [81]. In parallel, the housing condition induced changes in neuro-active small molecule/ metabolites in hippocampus and feces altered by the housing condition. Further analysis revealed that two major metabolites Benzenedicarboxylic acid and 2-Pregnane were detected in both feces and hippocampus. The estimated concentration of Benzenedicarboxylic acid was significantly higher in DW and EE mice in both feces and hippocampus, but lower in STSC and LTSC mice. The behavioural phenotype observed in the experimental groups can be associated with the estimated Benzenedicarboxylic acid in DW/EE mice [82] and 2-Pregnane in STSC and LTSC mice [83], which is precursor for corticosterone synthesis, possibly regulate the HPA axis [84] and interacted with the examined pathway in this study. Note to mention that we have used only male animals in this study partly based on the concern that oestrous-linked variations may occur in females. However, only little evidence is exist females make more various [85, 86], and arguing to use both sexes for any analysis. Therefore, further experiment will be performed in both sexes to understand the underlying mechanism.

## Conclusion

In the social living field mice *Mus booduga*, social isolation altered the synaptic plasticity, learning and memory. Subsequent exposure to an EE reversed the SI induced changes in the expression of genes associated with an ITSN1-Reelin-AMPA receptor pathway, and facilitates reference and working memory. The results observed in field mice *Mus booduga* are comparable to other laboratory bred rodent models and strengthen the mechanism in different rodent strains. Therefore, these findings support the view that EE could be an effective training mechanism for improving the memory against social stress-induced impairment.

## Supporting information

**S1 File.**
(DOC)

## Acknowledgments

We acknowledge the Department of Science and Technology (DST)-Fund for Improvement of S&T Infrastructure (FIST) for supporting Department of Animal Science.

## Author Contributions

**Conceptualization:** Koilmani Emmanuvel Rajan.

**Data curation:** Swamynathan Sowndharya.

**Formal analysis:** Swamynathan Sowndharya.

**Funding acquisition:** Koilmani Emmanuvel Rajan.

**Investigation:** Swamynathan Sowndharya, Koilmani Emmanuvel Rajan.

**Methodology:** Swamynathan Sowndharya.

**Project administration:** Koilmani Emmanuvel Rajan.

**Resources:** Koilmani Emmanuvel Rajan.

**Supervision:** Koilmani Emmanuvel Rajan.

**Validation:** Swamynathan Sowndharya.

**Writing – original draft:** Swamynathan Sowndharya.

**Writing – review & editing:** Koilmani Emmanuvel Rajan.

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
