## [Decision Letter · Decision Letter 0]

7 Sep 2023

PONE-D-23-13004Environmental enrichment improves social isolation-induced memory impairment:  The possible role of ITSN1-Reelin-AMPA receptor signaling pathwayPLOS ONE

Dear Dr. Rajan,

Thank you for submitting your manuscript to PLOS ONE. After careful consideration, we feel that it has merit but does not fully meet PLOS ONE’s publication criteria as it currently stands. Therefore, we invite you to submit a revised version of the manuscript that addresses the points raised during the review process.

We look forward to receiving your revised manuscript.

Kind regards,

Michelle Melgarejo da Rosa

Academic Editor

PLOS ONE

Journal Requirements:

2. In your Methods section, please provide additional information regarding the permits you obtained for the collection of the field-caught rodents. Please ensure you have included the full name of the authority that approved the process and, if no permits were required, a brief statement explaining why.

Please also include, in your Methods section, a statement confirming that the trapping method was specifically reviewed and approved by the IACUC

- https://www.sciencedirect.com/science/article/abs/pii/S0016648020303944?via%3Dihub

https://karger.com/dne/article-abstract/44/6/547/835246/Reduced-Reelin-Expression-Induces-Memory-Deficits?redirectedFrom=fulltext

https://journals.plos.org/plosone/article?id=10.1371%2Fjournal.pone.0127945

In your revision ensure you cite all your sources (including your own works), and quote or rephrase any duplicated text outside the methods section. Further consideration is dependent on these concerns being addressed

"The Department of Animal Science is supported by Department of Science and Technology (DST)-Fund for Improvement of S&T Infrastructure (FIST) (DST File No. SR/FST/LSI-647/2015(G) dt.11.08.2016), Rashtriya Uchchatar Shiksha Abhiyan (RUSA) 2.0-Biological Sciences (TN.No.311/RUSA/2018 dt.12.10.2020)."

"SS is recipient of Council of Scientific and Industrial Research (CSIR)-JRF Research Fellowship (File No. 09/475(0204)/2021-EMR-I). https://newfms.ncl.res.in/

KERKER received research grant from Department of Science & Technology (EMR/2016/005217 dt.21.03.2018)

https://www.serbonline.in/SERB/HomePage

6. Your ethics statement should only appear in the Methods section of your manuscript. If your ethics statement is written in any section besides the Methods, please move it to the Methods section and delete it from any other section. Please ensure that your ethics statement is included in your manuscript, as the ethics statement entered into the online submission form will not be published alongside your manuscript

7. In your Data Availability statement, you have not specified where the minimal data set underlying the results described in your manuscript can be found. PLOS defines a study's minimal data set as the underlying data used to reach the conclusions drawn in the manuscript and any additional data required to replicate the reported study findings in their entirety. All PLOS journals require that the minimal data set be made fully available. For more information about our data policy, please see http://journals.plos.org/plosone/s/data-availability.

8. Please upload a new copy of Figures 1 and 8 as the detail is not clear. Please follow the link for more information: " ext-link-type="uri" xlink:type="simple">https://blogs.plos.org/plos/2019/06/looking-good-tips-for-creating-your-plos-figures-graphics/"
https://blogs.plos.org/plos/2019/06/looking-good-tips-for-creating-your-plos-figures-graphics/

Reviewers' comments:

Reviewer's Responses to Questions

**Comments to the Author**

1. Is the manuscript technically sound, and do the data support the conclusions?

Reviewer #1: No

Reviewer #2: Yes

2. Has the statistical analysis been performed appropriately and rigorously? 

Reviewer #1: N/A

Reviewer #2: Yes

3. Have the authors made all data underlying the findings in their manuscript fully available?

Reviewer #1: No

Reviewer #2: Yes

4. Is the manuscript presented in an intelligible fashion and written in standard English?

Reviewer #1: No

Reviewer #2: No

5. Review Comments to the Author

**Reviewer #1: **A large body of literature has reported that enriched environments can ameliorate behavioral impairments caused by social isolation, including memory loss, so this study is less innovative. In addition, there are many defects in the Introduction, Methods, and Disscusssion, specific comments are as follows.

1. The writing quality of this manuscript needs to improve. For instance, in the Introduction Section, the authors need introduce the published work of social isolation-induced memory impairment as well as the ameliorating effect of environmental enrichment on social isolation-induced behavioral impairment. In the Discussion Section, the authors also need compare their finding with the previous published papers.

2. The description of animals in groups is not clear. How many animals in each group are kept in a cage and how many cages in each group? What principles are used to screen mice for behavioral, pathological, and molecular analysis? It is suggested that the author draw an experimental flow chart.

3. The resolution of the picture is too low. In particular, the symbols in the picture are difficult to distinguish. In addition, the layout of the pictures should be further optimized, such as Figure 3 and Figure 4 can be arranged in a page.

4. The statistical method, the meaning of the corresponding symbol in the figure, and the number of animals in each group should be indicated in the figure legends.

5. The authors also should make clear the reasons for the selection of male animals and for observing the hippocampus.

**Reviewer #2: **This study uses wild-caught mus booduga to investigate how environmental enrichment and social isolation alter many hippocampal molecular markers linked to synaptic transmission and neuronal excitability as well as impact cognitive performance. Overall, the experiments and well-performed, the analysis and statistics are robust. Some improvements are needed for the materials and methods, as well as improvements in figure readability. Importantly, correction needs to be made for the grammar errors throughout the text.

Please provide more details about the experimental methods, specifically the caging conditions:

- What bedding was used for all of the caging conditions?

- “How much is “plenty of food and water.” Ab libitum? Please specify

- “Healthy” mice were used. How much did they weigh? How was health assessed?

Please provide more detail about the scoring and analysis of the behavioral videos. How was the timing recorded? What software was used?

Please provide a definition of measures of working memory and reference memory in the methods.

Please provide title for the figures, this will enhance readability.

Figure 1 – Please provide explanation in the figure legend for the meaning of DW, STSC, LTSC and EE. Also, which test is being performed here?

Figure 2 – the bands are very dark- in the copy I have downloaded I can barely see the SYT4 and BDNF signals over the dark background.

Figure 4 – the background signal is very bright,, I cannot see the labelling well at all. Could higher-quality images please be provided? Also, the scale bars are illegible.

Figure 5 – same problem for figure 2.

Figure 8 – the text in panels a and b is illegible. TOO SMALL. Please adjust the figure so that normal sized font can be used.

There are many grammatical errors throughout the manuscript. I highly recommend using a grammar-checking program to improve the text.

6. PLOS authors have the option to publish the peer review history of their article (what does this mean?). If published, this will include your full peer review and any attached files.

Reviewer #1: No

Reviewer #2: No

---

## [Author Response · Author response to Decision Letter 0]

25 Oct 2023

Response Reviewers:

Reviewer #1: 

1. The writing quality of this manuscript needs to improve. For instance, in the Introduction Section, the authors need introduce the published work of social isolation-induced memory impairment as well as the ameliorating effect of environmental enrichment on social isolation-induced behavioral impairment. In the Discussion Section, the authors also need compare their finding with the previous published papers.

-Yes, in the revised manuscript we have included earlier reports in introduction section (Page No. 5, Line: 15-19) and also in the discussion section (Page No. 23, Line: 4-7 )

2. The description of animals in groups is not clear. How many animals in each group are kept in a cage and how many cages in each group?

- In the revised manuscript, the detailed number of animals (total number of animals captured; and final numbers taken for analysis is mentioned and number of animals housed in cage (Page No. 6, Line: 12; Page No. 7, Line: 6-16 )

 What principles are used to screen mice for behavioral, pathological, and molecular analysis? 

- In the revised manuscript we have included in the method section (Page No. 6, Line: 12-14; Page No. 7, Line: 18-19 )

It is suggested that the author draw an experimental flow chart.

- Yes, the flow chard with description included in the revised manuscript supplementary information. 

- 

3. The resolution of the picture is too low. In particular, the symbols in the picture are difficult to distinguish. In addition, the layout of the pictures should be further optimized, such as Figure 3 and Figure 4 can be arranged in a page.

- Yes, we have revised the figures as per the suggestion. 

- 

4. The statistical method, the meaning of the corresponding symbol in the figure, and the number of animals in each group should be indicated in the figure legends.

- Yes, we have increased the font size for symbol and included the number of animals in the figure legends of the revised the figures.

5. The authors also should make clear the reasons for the selection of male animals and for observing the hippocampus.

- Yes, we have included the statement in the discussion section of the revised manuscript (Page No. 27, Line: 16-20)

Reviewer #2: This study uses wild-caught mus booduga to investigate how environmental enrichment and social isolation alter many hippocampal molecular markers linked to synaptic transmission and neuronal excitability as well as impact cognitive performance. Overall, the experiments and well-performed, the analysis and statistics are robust. Some improvements are needed for the materials and methods, as well as improvements in figure readability. Importantly, correction needs to be made for the grammar errors throughout the text.

Please provide more details about the experimental methods, specifically the caging conditions:

- What bedding was used for all of the caging conditions?

-We have included in the details in the revised manuscript method section (Page No. 7, Line: 5-16)

“How much is “plenty of food and water.” Ab libitum? Please specify

We have used alternative to “ ad libitum” corrected in the revised manuscript (Page No. 6, Line: 15-16)

- “Healthy” mice were used. How much did they weigh? How was health assessed?

- In the revised manuscript we have included in the method section (Page No. 6, Line: 12-14; Page No. 7, Line: 18-19 )

Please provide more detail about the scoring and analysis of the behavioral videos. How was the timing recorded? What software was used?

- -We have included in the details in the revised manuscript method section (Page No. 8, Line: 16-17)

Please provide a definition of measures of working memory and reference memory in the methods.

- -We have included in the details in the revised manuscript method section (Page No. 8, Line: 1-3)

- 

Please provide title for the figures, this will enhance readability.

- Figure titles are included in the figure legend of the revised manuscript.

Figure 1 – Please provide explanation in the figure legend for the meaning of DW, STSC, LTSC and EE. Also, which test is being performed here?

- Details are included in the revised manuscript

Figure 2 – the bands are very dark- in the copy I have downloaded I can barely see the SYT4 and BDNF signals over the dark background.

- - Thank you, we have replaced the SYT4, BDNF, and other markers with new blots

- 

Figure 4 – the background signal is very bright,, I cannot see the labelling well at all. Could higher-quality images please be provided? Also, the scale bars are illegible.

- Thank you, the experiment was performed again and replaced the old images. 

- Figure 5 – same problem for figure 2.

- Thank you, we have replaced the dark background blot with new blots

Figure 8 – the text in panels a and b is illegible. TOO SMALL. Please adjust the figure so that normal sized font can be used.

- - Thank you, the larger font is used in the revised image.

---

## [Editor Report · Decision Letter 1]

31 Oct 2023

Environmental enrichment improves social isolation-induced memory impairment:  The possible role of ITSN1-Reelin-AMPA receptor signaling pathway

PONE-D-23-13004R1

Dear Dr. K.E. Rajan 

We’re pleased to inform you that your manuscript has been judged scientifically suitable for publication and will be formally accepted for publication once it meets all outstanding technical requirements.

Kind regards,

Michelle Melgarejo da Rosa

Academic Editor

PLOS ONE

---

## [Editor Report · Acceptance letter]

7 Nov 2023

PONE-D-23-13004R1 

Environmental enrichment improves social isolation-induced memory impairment:  The possible role of ITSN1-Reelin-AMPA receptor signaling pathway 

Dear Dr. Rajan:

I'm pleased to inform you that your manuscript has been deemed suitable for publication in PLOS ONE. Congratulations! Your manuscript is now with our production department. 

Kind regards, 

on behalf of

Dr. Michelle Melgarejo da Rosa 

Academic Editor

PLOS ONE